# Feasibility of local anesthesia for full-endoscopic spine surgery: A retrospective cohort study of body weight-based thresholds

Takafumi Ohshima [1,2], Takayuki Kitahara[1], Seiya Watanabe[1], Saori Soeda[1], Daiki Nakajima [1], Hiroshi Kageyama[1], Masatoshi Morimoto[1], Hiroaki Manabe[1], Fumitake Tezuka[1], Junzo Fujitani[1], Atsushi Teramoto [2], Koichi Sairyo[1]*

**1** Department of Orthopedics, University of Tokushima, Tokushima, Japan, **2** Department of Orthopaedics, Sapporo Medical University School of Medicine, Sapporo, Japan

* sairyokun@hotmail.com

## Abstract

### Background

While transforaminal full-endoscopic spine surgery (TF-FESS) under local anesthesia offers the advantages of real-time neurological feedback and minimized physiological stress, concerns persist regarding whether the procedure can be consistently completed within safe local anesthetic dosage limits in all patient populations. This study evaluated the feasibility and safety of TF-FESS under local anesthesia independent of frailty status using body weight-based safety thresholds for lidocaine.

### Methods

We retrospectively identified 108 patients who underwent single-level TF-FESS under local anesthesia and categorized them into a non-frail group (modified Frailty Index-5 [mFI-5] < 2, n = 80) and a frail group (mFI-5 ≥ 2, n = 28). Weight-based thresholds, calculated using actual body weight, were set at 7 mg/kg for non-frail patients and 5 mg/kg for frail patients. The anesthesia protocol consisted of sequential 1% lidocaine injection and conscious sedation. Complications were defined as clinically significant local anesthetic systemic toxicity, postoperative delirium, and permanent nerve root injury, with exclusion of transient intraoperative feedback.

### Results

Surgery was successfully completed under local anesthesia in all 108 cases (100%), with continuous verbal communication maintained throughout the procedure. The mean lidocaine dose was 218 ± 40 mg, with no significant difference between groups (p = 0.722). In the frail group, 92.9% of patients remained within the stricter 5 mg/ kg threshold. No documented cases of clinically significant local anesthetic systemic

**Data availability statement:** All relevant data are within the manuscript and its Supporting Information files.

**Funding:** The author(s) received no specific funding for this work.

**Competing interests:** The authors have declared that no competing interests exist.

toxicity, permanent nerve root injury, or postoperative delirium were observed in either group.

## Conclusions

TF-FESS under local anesthesia appears feasible, with no clinically significant complications observed in this cohort, irrespective of frailty status. Although the dosing thresholds were well-tolerated, they should be interpreted with caution given the study's retrospective nature. Adherence to weight-based dosing and a standardized conscious sedation protocol is associated with a favorable safety profile while preserving real-time neurological feedback. This strategy may minimize perioperative physiological stress and cognitive risks in an increasingly aging population.

## Introduction

One of the consequence of rapidly aging populations worldwide has been an increasing prevalence of degenerative spinal diseases requiring surgical intervention [1,2]. Multimorbidity and frailty are common among elderly patients, making conventional open surgery under general anesthesia a high-risk proposition because of the potential for perioperative complications, including cardiovascular events and postoperative delirium [3,4]. Transforaminal full-endoscopic spine surgery (TF-FESS) has become a leading minimally invasive spine surgery with significant advantages, including less tissue damage, minimal blood loss, and more rapid recovery [5,6].

While local anesthesia (LA) is often preferred for a transforaminal approach to allow real-time neurological feedback [7], general anesthesia remains widely used in many clinical settings to ensure patient comfort and procedural stability [8]. However, use of LA is gaining traction because it minimizes physiological stress and avoids general anesthesia-related risks, particularly in vulnerable populations such as the frail elderly [3,9]. The safety of LA is of particular concern in frail patients, whose decreased physiological reserve and potential variations in drug metabolism may narrow the therapeutic window for local anesthetic agents [10,11]. However, despite their clinical importance, empirical data comparing the safety and anesthetic requirements of TF-FESS under LA between frail and non-frail patients remain scarce.

The modified Frailty Index-5 (mFI-5) is a validated and efficient tool for assessment of frailty and predicting postoperative outcomes in various surgical specialties, including spine surgery [12,13]. Use of this index makes it possible to objectively categorize patients and evaluate the feasibility of TF-FESS under LA in high-risk groups. The purpose of this study was to evaluate the safety and clinical outcomes of TF-FESS under LA, specifically focusing on the comparison of local anesthetic dosage and complication rates between frail patients (mFI-5 ≥ 2) and non-frail patients (mFI-5 < 2). Our hypothesis was that TF-FESS under LA could be safely performed in frail patients without increasing the risk of clinically significant local anesthetic systemic toxicity (LAST) or other major complications.

## Materials and methods

### Ethics statement

The study was approved by the Institutional Review Board of Tokushima University Hospital (approval number 3642) and conducted in accordance with the principles expressed in the Declaration of Helsinki. Because of the retrospective nature of this study, the requirement for written informed consent was waived by the IRB. Instead, informed consent was obtained via the opt-out method, where information regarding the study was disclosed on the hospital's official website to provide patients with the opportunity to decline participation. All data were pseudonymized before analysis to ensure patient confidentiality. For research purposes, the medical records were accessed between 1 Dec 2025 and 15 Feb 2026. During and after data collection, the authors had no access to information that could identify individual participants.

### Study population and variables

We retrospectively identified all patients who underwent TF-FESS under LA at our institution between January 2024 and March 2025. Patients were excluded if they underwent surgeries involving two or more intervertebral levels or if the surgery was indicated for spinal infection.

Baseline demographic and surgical data for these patients were collected from the electronic medical records. The patient data collected included age, sex, actual body weight, relevant comorbidities (diabetes mellitus, hypertension, chronic obstructive pulmonary disease, and congestive heart failure), and functional status (evaluated as a categorical variable). Patient frailty was assessed using the mFI-5 and a corresponding binary frailty indicator [11,12].

Operative variables included the surgical diagnosis, operative level, operative time, and total dose of 1% lidocaine. To ensure the robustness of our safety analysis, specific complications were defined as follows:

### Local anesthetic systemic toxicity

Clinically significant LAST was assessed based on the occurrence of classic central nervous system symptoms (e.g., metallic taste, circumoral numbness, tinnitus, agitation, or seizures) or cardiovascular instability (e.g., bradycardia, arrhythmias, or hypotension) during or immediately after the procedure.

### Postoperative delirium

Postoperative delirium was defined as any documented episode of acute confusion, disorientation, or altered consciousness recorded in the electronic medical records by nursing or medical staff during the hospital stay. We note that standardized prospective tools, such as the Confusion Assessment Method (CAM), were not utilized in this retrospective review.

### Nerve root injury

Nerve root injury was defined strictly as new-onset postoperative motor weakness or sensory deficits that persisted beyond the immediate recovery period. Transient intraoperative leg pain or electric shock-like sensations during the procedure were considered essential components of real-time neurological feedback and were not classified as complications. Intraoperative tolerability measures included the need for intravenous rescue analgesics/sedatives and interruption/abandonment of the procedure as a result of pain.

### Anesthetic and sedation protocol

The LA procedure was performed by the primary surgeon before the surgical incision. Following administration of 1% lidocaine into the subcutaneous tissue and fascia, the anesthetic agent was sequentially injected into the facet joint capsule,

the inferior vertebral endplate, and the annulus fibrosus of the intervertebral disc. Injections were performed slowly to minimize the risk of rapid systemic absorption.

Vital signs, including peripheral oxygen saturation, electrocardiography, and non-invasive blood pressure, were monitored continuously throughout the procedure. Surgical nursing staff performed real-time clinical assessments for signs of LAST, and intravenous lipid emulsion was readily available in the operating room as a standard safety precaution.

A standardized conscious sedation protocol was used to ensure patient comfort while maintaining the capacity for neurological feedback. This consisted of intravenous administration of pentazocine (15 mg) and hydroxyzine pamoate (25 mg), which was divided into two doses and administered at 15-min intervals. This regimen allowed for an optimal balance between analgesia and preservation of real-time patient communication during the procedure.

### Definition of frailty groups

For subgroup analyses, patients were categorized based on their mFI-5 scores. The cohort was divided into a non-frail group (mFI-5 < 2) and a frail group (mFI-5 ≥ 2) for evaluation of the impact of physiological reserve on surgical and anesthetic outcomes.

### Safety threshold for local anesthesia

The individual safety limit for the non-frail group was defined as 7 mg/kg of actual body weight [3]. In contrast, the safety threshold for the frail group was set at a more conservative 5 mg/kg to account for their narrowed therapeutic window [3]. This lower threshold was established because frail patients often show increased drug absorption rates, reduced clearance, sarcopenia-related loss of drug reservoirs, and heightened pharmacological sensitivity [3,10]. The proportion of cases exceeding the respective dose limits was summarized for both the overall cohort and within each frailty group.

### Statistical analysis

Continuous variables are summarized as the mean ± standard deviation and categorical variables (including the proportion of cases exceeding the dose limit) as the number (percentage). Between-group comparisons (non-frail vs. frail) were performed using the Welch two-sample *t*-test for continuous variables and the chi-squared test or Fisher's exact test or Fisher's exact test for categorical variables as appropriate. Linear regression analysis was performed to visualize the relationship between actual body weight and the total lidocaine dose administered. The statistical analyses were performed using Python software (libraries including pandas, NumPy, and SciPy). All statistical tests were two-sided, and a p-value of < 0.05 was considered statistically significant. The minimal underlying dataset is provided as S1 data.

### Reporting guidelines

The study was conducted and reported in accordance with the STROBE (Strengthening the Reporting of Observational Studies in Epidemiology) guidelines for cohort studies.

## Results

### Study population and baseline characteristics

A total of 150 patients underwent TF-FESS during the study period. After 42 exclusions, (surgeries involving two or more intervertebral levels, n = 35; spinal infection, n = 7), the study cohort included 108 patients, with 80 (74.1%) categorized into the non-frail group (mFI-5 < 2) and 28 (25.9%) into the frail group (mFI-5 ≥ 2) (Fig 1). The mean age of the overall cohort was 61.9 ± 18.0 years, and 64 patients (59.3%) were male. The frail group was significantly older than the non-frail group (74.5 years ± 10.6 vs. 57.5 years ± 18.0 years, p < 0.001). There was no significant between-group difference in sex distribution (p = 0.685) or mean body weight (67.7 ± 13.1 kg vs. 66.8 ± 12.3 kg, p = 0.739).

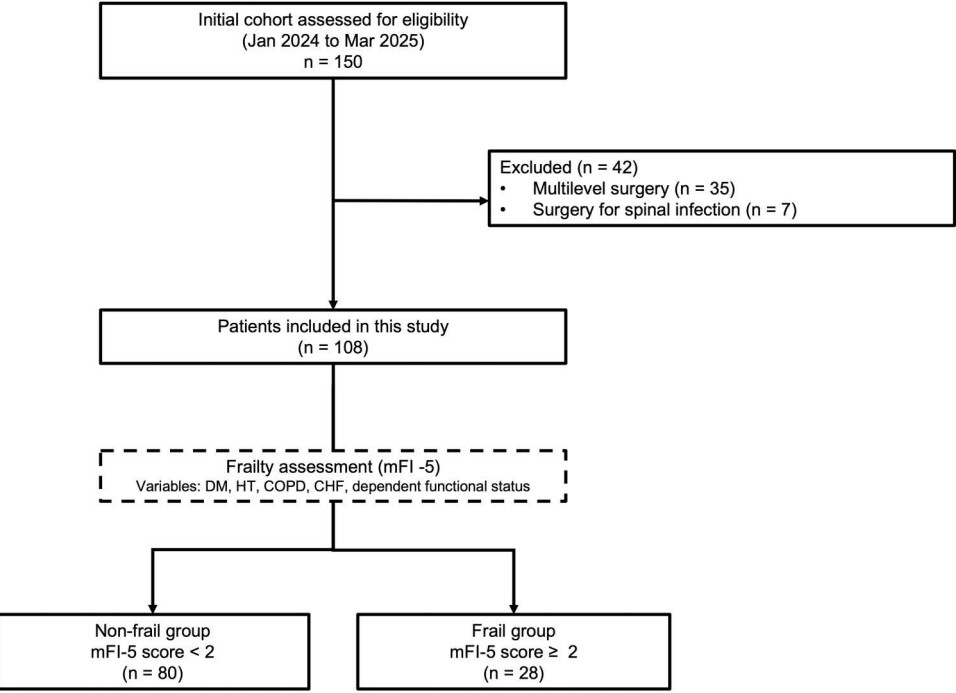

**Fig 1. Flow Diagram of Patient Selection.** In total, 108 of the 150 patients screened between January 2024 and March 2025 met the study inclusion criteria. Participants were categorized into a non-frail group (mFI-5 < 2, n = 80) and a frail group (mFI-5 ≥ 2, n = 28) based on the mFI-5 score. Abbreviations: CHF, congestive heart failure; COPD, chronic obstructive pulmonary disease; DM, diabetes mellitus; HT, hypertension; mFI-5, modified Frailty Index-5.

Consistent with the frailty classification, the prevalence of each of the comorbidities assessed was significantly higher in the frail group than in the non-frail group (diabetes mellitus, 60.7% vs. 6.2%, p < 0.001; hypertension, 78.6% vs. 21.2%, p < 0.001; chronic obstructive pulmonary disease, 14.3% vs. 1.2%, p = 0.016; chronic heart failure, 53.6% vs. 0.0%, p < 0.001; and dependent functional status (10.7% vs. 0.0%, p = 0.016). Furthermore, the estimated safety limit for LA was significantly lower in the frail group (334 ± 62 mg vs. 474 ± 92 mg, p < 0.001). The baseline demographic and clinical characteristics are summarized in Table 1.

Data are presented as the number (percentage) unless otherwise indicated. Abbreviations: COPD, chronic obstructive pulmonary disease; LA, local anesthesia; mFI-5, modified Frailty Index-5

### Surgical characteristics and intraoperative data

The surgical diagnoses primarily included lumbar disc herniation (65.7%), followed by lumbar lateral recess stenosis (17.6%) and lumbar foraminal stenosis (16.7%). The most frequently operated levels were L5/S1 (47.2%) and L4/5 (35.2%). There was no significant difference in the distribution of surgical diagnoses (p = 0.369) or the operative level (p = 0.125) between the non-frail and frail groups.

The mean operative time for the cohort overall was 83.8 ± 20.2 min, and estimated blood loss was minimal (4.6 ± 4.2 mL). Between-group comparisons found no significant difference in operative time between the non-frail and frail groups (84.5 ± 21.4 min vs. 81.6 ± 16.5 min, p = 0.467). Although the non-frail group had slightly greater estimated blood loss (5.0 ± 4.5 mL vs. 3.4 ± 3.1 mL), this difference reached only borderline significance (p = 0.050) and was not considered clinically meaningful. The surgical data are detailed in Table 2.

**Table 1. Patient demographics and baseline characteristics.**

| Variable | Total cohort (n = 108) | Non-frail group (n = 80) | Frail group (n = 28) | p-value |
|---|---|---|---|---|
| **Demographic data** | | | | |
| Age (years) | 61.9 ± 18.0 | 57.5 ± 18.0 | 74.5 ± 10.6 | <0.001 |
| Male sex | 64 (59.3%) | 46 (57.5%) | 18 (64.3%) | 0.685 |
| Weight (kg) | 67.4 ± 12.9 | 67.7 ± 13.1 | 66.8 ± 12.3 | 0.739 |
| Estimated limit of LA (mg) | 438 ± 105 | 474 ± 92 | 333.8 ± 62 | <0.001 |
| **Comorbidity** | | | | |
| Diabetes mellitus | 22 (20.4%) | 5 (6.2%) | 17 (60.7%) | <0.001 |
| Hypertension | 39 (36.1%) | 17 (21.2%) | 22 (78.6%) | <0.001 |
| COPD | 5 (4.6%) | 1 (1.2%) | 4 (14.3%) | 0.016 |
| Chronic heart failure | 15 (13.9%) | 0 (0.0%) | 15 (53.6%) | <0.001 |
| Dependent functional status | 3 (2.8%) | 0 (0.0%) | 3 (10.7%) | 0.016 |
| mFI-5 score | | | | <0.001 |
| 0 | 57 (52.8%) | 57 (71.2%) | 0 (0.0%) | |
| 1 | 23 (21.3%) | 23 (28.8%) | 0 (0.0%) | |
| 2 | 23 (21.3%) | 0 (0.0%) | 23 (82.1%) | |
| 3 | 5 (4.6%) | 0 (0.0%) | 5 (17.9%) | |
| 4 | 0 (0.0%) | 0 (0.0%) | 0 (0.0%) | |
| 5 | 0 (0.0%) | 0 (0.0%) | 0 (0.0%) | |

Data are presented as the mean ± standard deviation or as the number (percentage) as appropriate. Abbreviation: LAST, local anesthesia systemic toxicity

## Safety and outcomes of local anesthesia

In all 108 cases (100%), continuous verbal communication with the patient was successfully maintained throughout the procedure, and the surgery was completed under LA without any need for conversion to general anesthesia. The mean total dose of lidocaine administered intraoperatively was 218 ± 40 mg, with no significant difference observed between the non-frail and frail groups (219 ± 41 mg vs. 216 ± 37 mg, p = 0.722). Evaluation of the total administered volume against the prespecified weight-based safety limits (7 mg/kg for non-frail patients and 5 mg/kg for frail patients) identified only 2 patients (1.9%) in the overall cohort in whom the respective thresholds were exceeded. Subgroup analysis revealed that no patients (0.0%) in the non-frail group exceeded the limit, whereas 2 patients (7.1%) in the frail group exceeded the stricter 5 mg/kg threshold (Fig 2). This difference did not reach statistical significance (p = 0.065). Linear regression analysis demonstrated that lidocaine dosing trends were nearly identical between the non-frail and frail groups. While the regression lines remained generally below the institutional thresholds for the majority of the cohort, the frail regression line converged toward and slightly intersected the 5 mg/kg limit in the lowest body weight range (Fig 2).

Importantly, the procedure had an excellent safety profile. Regardless of frailty status and whether or not the LA safety threshold was exceeded, there were no documented cases of clinically significant LAST symptoms, nerve root injury, postoperative delirium, or conversion to general anesthesia across the entire cohort (p > 0.999 for all complications). No intravenous rescue analgesics/sedatives were required, and no procedures were interrupted or abandoned because of intolerable pain. Given the zero-event nature of these outcomes, the upper bound of the 95% confidence interval for the event rate was approximately 3.4% for the overall cohort (0/108) and 12.3% for the frail subgroup (0/28).

**Table 2. Surgical characteristics and operative outcomes.**

| | Total cohort (n = 108) | Non-frail group (n = 80) | Frail group (n = 28) | p-value |
|---|---|---|---|---|
| **Surgical data** | | | | |
| Diagnosis | | | | 0.369 |
| Lumbar disc herniation | 71 (65.7%) | 53 (66.2%) | 18 (64.3%) | |
| Lumbar foraminal stenosis | 18 (16.7%) | 15 (18.8%) | 3 (10.7%) | |
| Lumbar lateral recess stenosis | 19 (17.6%) | 12 (15.0%) | 7 (25.0%) | |
| Operated level | | | | 0.125 |
| L2/3 | 6 (5.6%) | 2 (2.5%) | 4 (14.3%) | |
| L3/4 | 13 (12.0%) | 10 (12.5%) | 3 (10.7%) | |
| L4/5 | 38 (35.2%) | 28 (35.0%) | 10 (35.7%) | |
| L5/S1 | 51 (47.2%) | 40 (50.0%) | 11 (39.3%) | |
| **Intraoperative data** | | | | |
| Operative time (min) | 83.8±20.2 | 84.5±21.4 | 81.6±16.5 | 0.467 |
| Blood loss (mL) | 4.6±4.2 | 5.0±4.5 | 3.4±3.1 | 0.05 |
| Local anesthesia, lidocaine | | | | |
| Total volume (mg) | 218±40 | 219±41 | 216±37 | 0.722 |
| Safety limit Exceeded | 2 (1.9%) | 0 (0.0%) | 2 (7.1%) | 0.065 |
| **Complications** | | | | >0.999 |
| LAST symptoms | 0 (0.0%) | 0 (0.0%) | 0 (0.0%) | |
| Nerve root injury | 0 (0.0%) | 0 (0.0%) | 0 (0.0%) | |
| Postoperative delirium | 0 (0.0%) | 0 (0.0%) | 0 (0.0%) | |
| Conversion to general anesthesia | 0 (0.0%) | 0 (0.0%) | 0 (0.0%) | |

## Discussion

This study showed that TF-FESS under LA was feasible in this cohort, including patients categorized as frail. Notably, across our entire cohort of 108 patients, there were no documented instances of clinically significant LAST, postoperative delirium, permanent nerve root injury, or need for conversion to general anesthesia. These findings suggest that our weight-based LA dosing protocol used I this study was tolerated within this cohort irrespective of frailty status, although the retrospective design and absence of serum lidocaine measurements warrant cautious interpretation.

### Safety of local anesthesia and weight-based dosing thresholds

A primary concern during TF-FESS under LA is the risk of LAST, particularly in elderly or frail patients, in whom drug metabolism is potentially altered [3,10,14]. To address this concern, we implemented weight-based safety thresholds, namely, 7 mg/kg for the non-frail group and a more conservative 5 mg/kg for the frail group calculated based on actual body weight. While these thresholds exceed the 4.5 mg/kg limit for plain lidocaine specified in the U.S. Food and Drug Administration (FDA)-approved labeling, our clinical observations suggest they are well-tolerated in the specific context of TF-FESS [3,10,15]. In the non-frail group (n = 80), 100% of patients remained within their safety limit. In the frail group (n = 28), the 5 mg/kg threshold was complied with in 92.9% of cases; although this limit was exceeded slightly in two frail cases because of operative requirements, no clinical symptoms of toxicity were observed. This consistent safety profile across both groups could potentially be explained by the specific anatomical and vascular characteristics of the surgical site. It is hypothesized that the systemic absorption of lidocaine from the subcutaneous tissue and fascia is relatively slow. Furthermore, a substantial volume of lidocaine was administered into the facet joint capsule. The dense fibrous

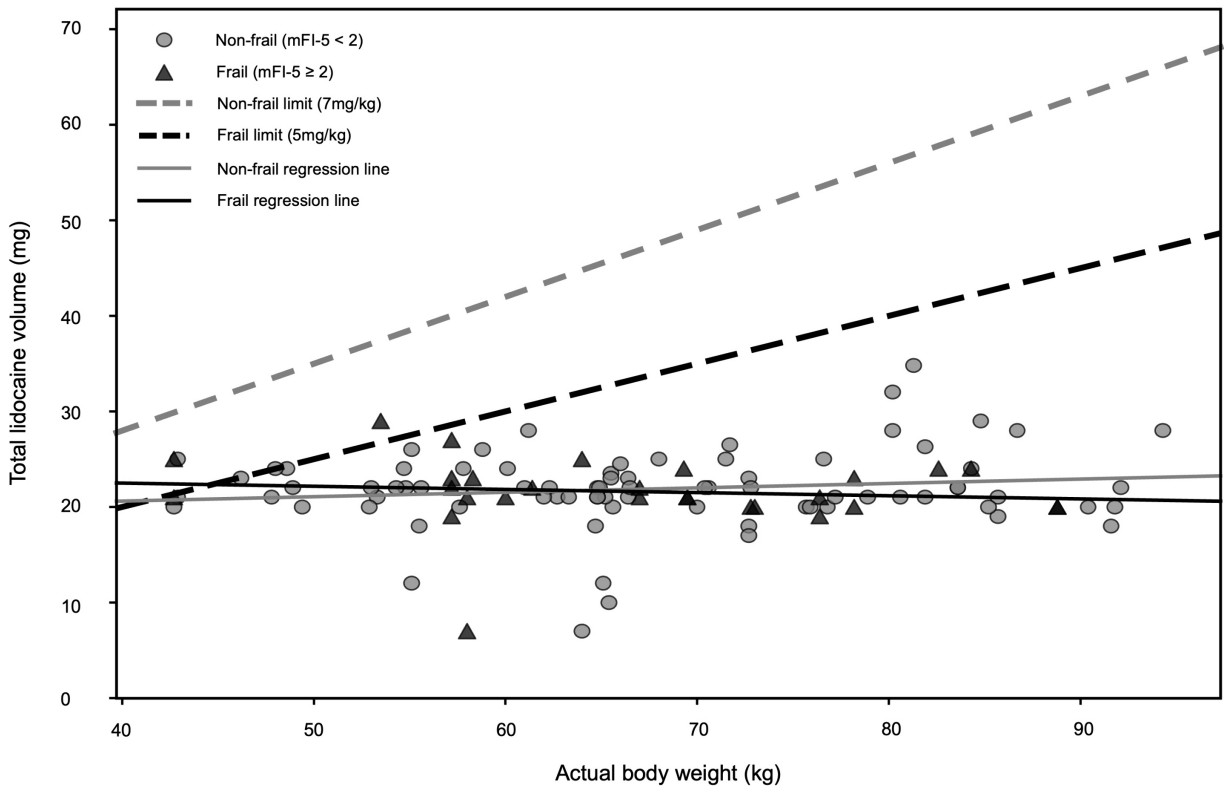

**Fig 2. Relationship Between Total Lidocaine Dosage and Actual Body Weight Relative to Frailty-Adjusted Safety Limits (n = 108).** The individual scatter plots represent the total dose of lidocaine (mg) administered during the procedure. The dashed gray line indicates the conventional safety limit for non-frail patients (7 mg/kg), while the dashed black line represents the stricter safety threshold (5 mg/kg) applied to frail patients (mFI-5 ≥ 2). The thin solid lines represent the linear regression of actual doses for each group. Gray circles denote non-frail patients (mFI-5 < 2), and black triangles denote frail patients. Two frail cases slightly exceeded the 5 mg/kg threshold; no local anesthesia systemic toxicity symptoms were observed in these cases. Abbreviation: mFI-5, modified Frailty Index-5.

structure of the joint capsule is hypothesized to act as a structural barrier, potentially sequestering part of the anesthetic agent within the intra-articular space and limiting its rapid diffusion to the surrounding extra-capsular tissues, which are more highly vascularized. By restricting immediate access to the systemic circulation in this manner, the joint capsule may help mitigate a rapid increase in intravascular lidocaine concentrations, potentially resulting a more gradual systemic uptake. This, combined with the poorly vascularized nature of the vertebral endplates and intradiscal spaces might further limit rapid systemic entry, although pharmacological studies measuring serum concentrations are required to confirm this mechanism [10,15]. Our data suggest that these weight-based thresholds may serve as a useful clinical benchmark for minimizing the risk of clinically significant LAST in patients undergoing TF-FESS under LA.

However, the interpretation of the 0% incidence of LAST in this study should be cautious. The retrospective design and reliance on medical record documentation primarily reflect the absence of documented, clinically significant LAST requiring medical intervention rather than the complete absence of all LAST-related symptoms. Additionally, the conscious sedation protocol using pentazocine and hydroxyzine could have potentially masked early, mild neurological signs of toxicity, such as metallic taste or circumoral numbness, which might have skewed the reported safety data.

Regarding the frailty spectrum, it is important to note that our frail group (mFI-5 ≥ 2) predominantly consisted of patients with mild frailty (mFI-5 = 2, 82.1%). The study lacked patients with severe frailty (mFI-5 ≥ 4), which may limit the

generalizability of our findings to the most vulnerable populations. However, it is noteworthy that we applied the same conservative 5 mg/kg safety threshold to these mildly frail patients as would be used for more severely frail individuals. The fact that the vast majority of these patients were successfully managed within this stricter limit, with dosing trends remaining consistent and generally below the threshold. Further underscores the safety margin of our protocol. Nevertheless, our results highlight that TF-FESS under LA is a viable and low-stress alternative for mildly frail patients who might otherwise face higher risks under general anesthesia.

### Clinical safety: Prevention of postoperative delirium and nerve root injury

The complete absence of documented postoperative delirium and nerve root injury across our entire cohort of 108 patients represents a noteworthy clinical observation. Postoperative delirium is a common and distressing complication in elderly patients undergoing spinal surgery under general anesthesia [4]. As shown in Table 3, Tacconi et al. reported a 14.5% postoperative delirium rate under general anesthesia in an elderly cohort, whereas no cases occurred under spinal anesthesia [16]. Although direct comparison is limited by differences in patient age and surgical techniques among these studies, our findings suggest that TF-FESS under LA may offer similar potential advantages in a broader surgical population. The retrospective identification of delirium through medical records without standardized tools like the Confusion Assessment Method may have under-detected hypoactive cases. By mitigating the physiological stress associated with general anesthesia and maintaining the patient's conscious state, this approach suggests a potential for minimizing the risk of postoperative cognitive decline, even in individuals with frailty.

Furthermore, the 0% nerve root injury rate suggests that TF-FESS under LA can be performed with a favorable safety profile. In contrast with historical data for endoscopic surgery under general anesthesia, where injury rates reached as high as 14.6% even in younger populations (Table 3) [17,18], use of LA allowed for continuous real-time neurological feedback from the patient [7,8]. This "biological monitoring" was effective regardless of frailty status, ensuring that any proximity to neural structures was immediately communicated to the surgeon [7,8,19]. The consistency of these results across our entire cohort underscores the potential of TF-FESS under LA not only as a viable alternative for high-risk patients but also as a valuable strategy for improving procedural safety in degenerative spinal surgery.

### Limitations

This study has several limitations. First, it had a retrospective, single-center design and a relatively small sample size in the frail group (n = 28), which primarily consisted of mildly frail patients. While no major complications were observed, larger multicenter prospective studies are needed to confirm the generalizability of our findings. Second, we did not measure serum lidocaine concentrations; therefore, our assessment of local anesthetic safety remains based on clinical observations. Third, the possibility of under-detection of postoperative delirium and mild LAST symptoms as a consequence of

**Table 3. Selected reports of perioperative complications in patients who underwent lumbar spine surgery.**

| Study | Procedure | Anesthesia method | N | Mean age, years | Postoperative delirium rate | Nerve root injury rate |
|---|---|---|---|---|---|---|
| Present study | FESS | LA | 108 | 61.9 | 0% (0.0-3.4%) | 0% (0.0-3.4%) |
| Wu et al. [18] | PELD | GA | 36 | 48.8 | – | 8.30% |
| Ren et al. [19] | PELD | GA | 41 | 48.8 | – | 14.60% |
| Tacconi et al. [17] | PELD | GA vs SA | 27 vs 48 | 79.6 | 14.5% vs 0.0% | 8% |

Values for the present study are presented as percentage (95% confidence interval).

Abbreviations: FESS, full-endoscopic spine surgery; GA, general anesthesia; LA, local anesthesia; PELD, percutaneous endoscopic lumbar discectomy; SA, spinal anesthesia.

the retrospective nature of the research cannot be excluded. Given that these outcomes were identified by review of medical records rather than standardized prospective screening tools, such as the Confusion Assessment Method, subclinical or hypoactive delirium and transient systemic toxicity might have been overlooked. However, it is important to note that no clinically significant episodes (i.e., those requiring medical intervention, causing safety concerns, or impacting the clinical course) were documented throughout the study period. Moreover, mild central nervous system symptoms of LAST could have been partially masked by the effects of the conscious sedation protocol. Fourth, the long-term clinical outcomes beyond the perioperative period were not evaluated. Future research should compare the long-term functional recovery and mortality rates between TF-FESS under LA and TF-FESS under general anesthesia in the frail population.

## Conclusion

TF-FESS under LA appears to be a safe and feasible surgical option within this study cohort for both frail and non-frail patients. This is supported by the absence of documented clinically significant LAST, postoperative delirium, and permanent nerve root injury in our study population. Adherence with a weight-based LA dosing protocol, specifically a conservative 5 mg/kg threshold calculated based on actual body weight for frail individuals, is associated with a favorable safety profile, even in patients with multiple comorbidities. Avoiding general anesthesia and using real-time neurological feedback may help to minimize perioperative physiological stress and potentially enhance procedural safety. The findings of this study suggest that TF-FESS under LA may represent a potentially valuable and reliable strategy for managing degenerative spinal diseases in an increasingly aging and high-risk population.

## Supporting information

**S1 data. Minimal underlying dataset.This excel file contains the anonymized raw clinical data used for the analysis in this study.**
(XLSX)

## Author contributions

**Conceptualization:** Takafumi Ohshima, Koichi Sairyo.

**Data curation:** Takafumi Ohshima.

**Formal analysis:** Takafumi Ohshima, Atsushi Teramoto.

**Investigation:** Takafumi Ohshima.

**Project administration:** Takafumi Ohshima, Koichi Sairyo.

**Visualization:** Takafumi Ohshima.

**Writing – original draft:** Takafumi Ohshima.

**Writing – review & editing:** Takayuki Kitahara, Seiya Watanabe, Saori Soeda, Daiki Nakajima, Hiroshi Kageyama, Masatoshi Morimoto, Hiroaki Manabe, Fumitake Tezuka, Junzo Fujitani, Atsushi Teramoto.

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
