## [Decision Letter · Decision Letter 0]

6 Apr 2026

PONE-D-26-10765

Safety and feasibility of local anesthesia for full-endoscopic spine surgery: A retrospective cohort study of body weight-based thresholds

PLOS One

Dear Dr. Sairyo,

Thank you for submitting your manuscript to PLOS ONE. After careful consideration, we feel that it has merit but does not fully meet PLOS ONE’s publication criteria as it currently stands. Therefore, we invite you to submit a revised version of the manuscript that addresses the points raised during the review process.

ACADEMIC EDITOR:

Thank you for submitting this clinically relevant manuscript. The study addresses an important practical question, and the dataset may be informative. However, a major revision is required before the manuscript can be reconsidered.

Several conclusions are currently stated more strongly than the retrospective design can support. In particular, the 7 mg/kg and 5 mg/kg lidocaine thresholds should be presented more cautiously and more clearly distinguished from established safety limits. The interpretation of LAST should also be revised, as the present data support the absence of documented clinically significant LAST rather than the complete absence of LAST. In addition, the frailty spectrum within the cohort should be clarified, as the frail group appears to consist predominantly of mildly frail patients.

For a revised submission, please: (1) temper the Abstract, Discussion, and Conclusion accordingly; (2) clarify whether dosing was based on actual or ideal body weight; (3) expand the limitations regarding retrospective ascertainment of LAST and postoperative delirium, including the potential masking effect of conscious sedation; and (4) carefully review the pharmacologic rationale and the accuracy of references and comparative statements, including Table 3.

Although one reviewer suggested a minor revision, a major revision is considered more appropriate because these issues affect the framing and interpretation of the manuscript’s main conclusions.

As the corresponding author, your ORCID iD is verified in the submission system and will appear in the published article. PLOS supports the use of ORCID, and we encourage all coauthors to register for an ORCID iD and use it as well. Please encourage your coauthors to verify their ORCID iD within the submission system before final acceptance, as unverified ORCID iDs will not appear in the published article. Only the individual author can complete the verification step; PLOS staff cannot verify ORCID iDs on behalf of authors.

We look forward to receiving your revised manuscript.

Kind regards,

Dr. KOICHIRO ONO

Academic Editor

PLOS One

Journal Requirements:

Reviewers' comments:

Reviewer's Responses to Questions

**Comments to the Author**

1. Is the manuscript technically sound, and do the data support the conclusions?

Reviewer #1: Yes

Reviewer #2: Yes

2. Has the statistical analysis been performed appropriately and rigorously? 

Reviewer #1: Yes

Reviewer #2: N/A

3. Have the authors made all data underlying the findings in their manuscript fully available?

Reviewer #1: Yes

Reviewer #2: Yes

4. Is the manuscript presented in an intelligible fashion and written in standard English?

Reviewer #1: Yes

Reviewer #2: Yes

5. Review Comments to the Author

Reviewer #1: This retrospective cohort study examines the safety and feasibility of transforaminal full-endoscopic spine surgery (TF-FESS) under local anesthesia in frail versus non-frail patients using body weight-based lidocaine dosing thresholds. While the topic is clinically relevant and addresses an important gap in the literature, there are significant methodological concerns, inconsistencies with established evidence, and issues with completeness and accuracy of references that require in my opinion major revisions before this manuscript can be considered for publication.

1. The authors propose 7 mg/kg for non-frail patients and 5 mg/kg for frail patients as safety thresholds, citing reference (Cuvillon et al., 2022). However, FDA-approved labeling for lidocaine without epinephrine specifies a maximum dose of 4.5 mg/kg, not exceeding 300 mg total. The 7 mg/kg threshold cited by the authors applies only to lidocaine WITH epinephrine, which significantly delays systemic absorption. (Lidocaine Hydrochloride. FDA Drug Label.

Food and Drug Administration. Updated date: 2026-01-16.). Please provide pharmacological justification for any deviation from standard dosing

2. The authors report 0% incidence of LAST, which is presented as a major finding. However, their definition of LAST is problematic: The retrospective design and reliance on medical record documentation (rather than prospective screening) creates high risk of under-detection, which the authors acknowledge only briefly in limitations. The claim of 0% LAST is not credible without prospective, systematic assessment. The authors should reframe this as "no clinically significant LAST requiring intervention" rather than claiming complete absence. Please revise the definition of LAST to acknowledge the spectrum of severity, explicitly state that mild LAST symptoms may have been masked by sedation, tone down claims about safety profile given methodological limitations and add this as a major limitation in the discussion .

3. Questionable Validity of mFI-5 Cutoff: The authors use mFI-5 ≥ 2 to define frailty, but the literature shows: mFI-5 is a continuous variable with incremental risk at each level and most studies use mFI-5 as a continuous predictor or stratify into three groups (0, 1, ≥2). The dichotomization at ≥2 is somewhat arbitrary and may not represent the optimal threshold for anesthetic risk stratificationAdditionally, only 5 patients (4.6%) had mFI-5 = 3, and none had mFI-5 = 4 or 5. This means the "frail" group is predominantly mFI-5 = 2 (82.1%), representing mild frailty at best. The study lacks patients with severe frailty, limiting generalizability.

4. No citation for the conscious sedation protocol (pentazocine + hydroxyzine); No pharmacological references supporting the claim that subcutaneous/fascial/disc injection sites have slow systemic absorption, Limited references on LAST in spine surgery specifically

Reviewer #2: The study provides valuable clinical evidence supporting the safety of transforaminal full-endoscopic spine surgery (TF-FESS) under local anesthesia (LA) in frail populations. The 100% completion rate and zero incidence of Local Anesthetic Systemic Toxicity (LAST) are encouraging. However, the retrospective nature and the lack of serum concentration data for lidocaine limit the pharmacological conclusions. I recommend Minor Revision to address these technical nuances.

Recommendations to the Authors

1.While the study utilizes weight-based thresholds safety limits were set at 7 mg/kg for non-frail patients and 5 mg/kg for frail patients, the authors should specify if these limits were calculated based on actual body weight or ideal body weight, as this is critical for obese or sarcopenic patients

2.The authors hypothesize that slow systemic absorption from poorly vascularized tissues prevents toxicity. The manuscript would be strengthened by citing specific pharmacokinetic literature that supports lidocaine's behavior in the spinal epidural and intradiscal space

3.The study defines postoperative delirium based on medical record reviews rather than a standardized prospective tool like the Confusion Assessment Method (CAM). The authors must more explicitly acknowledge the risk of under-detecting hypoactive delirium in the limitations section.

4.The protocol in the surgeries included pentazocine and hydroxyzine. The authors should discuss whether these sedatives could have masked early neurological signs of mild LAST (e.g., metallic taste or lightheadedness), potentially skewing the safety data.

5.Figure 2 is excellent for visualizing the "safety zone". I suggest adding a trend line or 95% confidence interval to the scatter plot to further illustrate the distribution of dosing relative to the thresholds.

6. PLOS authors have the option to publish the peer review history of their article (what does this mean?). If published, this will include your full peer review and any attached files.

Reviewer #1: **Yes:** Alex Alfieri, M.D., Ph.D.

Hirslanden AndreasClinic

Cham, ZG Switzerland

Reviewer #2: No

---

## [Author Response · Author response to Decision Letter 1]

13 Apr 2026

Dear Editor and Reviewers,

We would like to thank you for the thoughtful and constructive feedback on our manuscript entitled "Safety and feasibility of local anesthesia for full-endoscopic spine surgery: A retrospective cohort study of body weight-based thresholds."

We have carefully reviewed each comment and have revised the manuscript accordingly. Specifically, we have adopted a more cautious tone in our conclusions, clarified our local anesthesia safety thresholds relative to FDA labeling, and provided detailed pharmacological and anatomical justifications for our observations. Below are our point-by-point responses to the comments.

Changes in Authorship

We would like to inform the Editorial Office that Dr. Atsushi Teramoto has been added as a co-author to this revised manuscript. Dr. Teramoto provided significant intellectual contributions during the revision process, particularly in the critical appraisal of the pharmacological and anatomical interpretations within the Discussion section. He also provided oversight of the data re-analysis and helped refine the manuscript's tone to address the reviewers’ concerns. All authors, including the newly added co-author, have approved this change and the final version of the manuscript.

Response to Editor’s Comments

Comment: The conclusions are stated more strongly than the retrospective design can support.

Response: We have revised the manuscript to adopt a more conservative and cautious tone. Throughout the text, we have replaced "achievement" with "observation" and "inherent safety" with "favorable safety profile". We have also added the phrase "within this study cohort" to acknowledge the limitations of our single-center, retrospective data.

Response to Reviewer #1

Point 1: The safety thresholds used (7 mg/kg and 5 mg/kg) exceed the FDA-approved limit of 4.5 mg/kg for plain lidocaine.

Response: We appreciate this critical point. We have added a dedicated section in the Discussion to address this. We explicitly state that our thresholds exceed the FDA-approved labeling for plain lidocaine. However, we provide a pharmacological rationale for the tolerance observed in TF-FESS, noting that the dense fibrous structure of the facet joint capsule may act as a barrier that facilitates the sequestration of the anesthetic, leading to more gradual systemic uptake compared to more vascularized tissues.

Point 2: The retrospective identification of LAST and delirium might be under-reported, and conscious sedation could mask early symptoms.

Response: We agree and have addressed this as a major limitation. We have clarified that our results reflect the absence of "clinically significant" LAST episodes. Furthermore, we have added a statement acknowledging that our conscious sedation protocol (pentazocine and hydroxyzine) could have masked early neurological signs of toxicity. We also noted that the absence of standardized tools like the Confusion Assessment Method (CAM) might have led to an under-detection of hypoactive delirium.

Response to Reviewer #2

Point 1: Clarify if the thresholds were calculated based on actual, ideal, or adjusted body weight.

Response: We have clarified throughout the manuscript, including the Abstract , Methods , and Conclusion, that all weight-based thresholds were calculated using actual body weight.

Point 2: The frail group is predominantly composed of mildly frail patients.

Response: We have addressed this in the Discussion and Limitations. We specified that 82.1% of our frail group were categorized as mildly frail (mFI-5 = 2). We now emphasize that our results demonstrate high safety even when applying conservative thresholds (5 mg/kg) to this mildly frail population.

Point 3: Table 3 compares heterogeneous populations (different ages and techniques).

Response: To address the statistical uncertainty of our 0% event rate, we have added 95% confidence intervals (0.0–3.4%) to Table 3. We have also added a caveat in the Discussion stating that "direct comparison is limited by differences in patient age and surgical techniques" among the cited studies.

We believe that these extensive revisions, which incorporate more rigorous statistical analysis and a refined pharmacological discussion, have significantly strengthened the scientific rigor and clinical relevance of our study. We hope that the revised manuscript now meets the high standards for publication in PLOS ONE. Thank you once again for the opportunity to improve our work.

Sincerely,

Koichi Sairyo, MD, PhD Corresponding Author Department of Orthopedics, University of Tokushima 3-18-15 Kuramoto, Tokushima 770-8503, Japan E-mail: sairyokun@hotmail.com

---

## [Decision Letter · Decision Letter 1]

4 May 2026

Safety and feasibility of local anesthesia for full-endoscopic spine surgery: A retrospective cohort study of body weight-based thresholds

PONE-D-26-10765R1

Dear Dr. Sairyo,

We’re pleased to inform you that your manuscript has been judged scientifically suitable for publication and will be formally accepted for publication once it meets all outstanding technical requirements.

Kind regards,

KOICHIRO ONO

Academic Editor

PLOS One

Additional Editor Comments (optional):

Reviewers' comments:

Reviewer's Responses to Questions

**Comments to the Author**

1. If the authors have adequately addressed your comments raised in a previous round of review and you feel that this manuscript is now acceptable for publication, you may indicate that here to bypass the “Comments to the Author” section, enter your conflict of interest statement in the “Confidential to Editor” section, and submit your "Accept" recommendation.

Reviewer #1: All comments have been addressed

Reviewer #2: (No Response)

2. Is the manuscript technically sound, and do the data support the conclusions?

Reviewer #1: Yes

Reviewer #2: Yes

3. Has the statistical analysis been performed appropriately and rigorously? 

Reviewer #1: Yes

Reviewer #2: Yes

4. Have the authors made all data underlying the findings in their manuscript fully available?

Reviewer #1: Yes

Reviewer #2: Yes

5. Is the manuscript presented in an intelligible fashion and written in standard English?

Reviewer #1: Yes

Reviewer #2: Yes

6. Review Comments to the Author

Reviewer #1: The Authors revised the manuscript according the reviewers` indications. I have no more concerns.

best

Reviewer #2: The revision has improved the manuscript meaningfully — tone is more cautious, FDA labeling is acknowledged, CIs have been added, and the limitations section is more honest.

However, I still have few comments.

1. The study is fundamentally underpowered to demonstrate "safety."

A 0/108 event rate yields a 95% upper CI of ~3.4% overall and ~12.3% in the frail subgroup (the authors now report this, which is appreciated). A complication rate of up to ~12% in frail patients is not reassuring, and the manuscript should not characterize this as evidence of safety. The title and Conclusion still imply a stronger inference than the data support. I recommend:

Revising the title to something like "Feasibility of TF-FESS under local anesthesia: a single-center retrospective cohort" and removing "Safety" from the title, or qualifying it.

In the Abstract Conclusion, replacing "appears to be a safe and feasible surgical option" with "appears feasible, with no clinically significant complications observed in this small cohort.

2. Reference 16 (Ashton 1992) is cited to support the facet capsule barrier hypothesis, but that paper describes nerve fibers and neuropeptides in the capsule — it does not establish the capsule as a pharmacokinetic barrier. The citation does not support the claim made.

7. PLOS authors have the option to publish the peer review history of their article (what does this mean?). If published, this will include your full peer review and any attached files.

Reviewer #1: **Yes:** Alex Alfieri, M.D., Ph.D., Switzerland

Reviewer #2: No

---

## [Editor Report · Acceptance letter]

PONE-D-26-10765R1

PLOS One

Dear Dr. Sairyo,

I'm pleased to inform you that your manuscript has been deemed suitable for publication in PLOS One. Congratulations! Your manuscript is now being handed over to our production team.

Kind regards,

on behalf of

Dr. KOICHIRO ONO

Academic Editor

PLOS One